# Aged gut microbiota contribute to different changes in antioxidant defense in the heart and liver after transfer to germ-free mice

**Yang Hong, Han Dong, Jing Zhou, Ya Luo, Ming-Ming Yuan, Jia-Fei Zhan, Yang-Lu Liu, Jie-Ying Xia, Lei Zhang**[ID]*

Animal Experiment Center of Sichuan Academy of Traditional Chinese Medicine Sciences, Chengdu, China

* leizhangTCM@163.com

**Data Availability Statement:** All data generated or analyzed during this study are included in this article.

## Abstract

Age-associated impairment in antioxidant defense is an important cause of oxidative stress, and elderly individuals are usually associated with gut microbiota (GM) changes. Studies have suggested a potential relationship between the GM and changes in antioxidant defense in aging animals. Direct evidence regarding the impact of aging-associated shifts in GM on the antioxidant defense is lacking. The heart is a kind of postmitotic tissue, which is more prone to oxidative stress than the liver (mitotic tissue). To test and compare the influence of an aged GM on antioxidant defense changes in the heart and liver of the host, in this study, GM from young adolescent (5 weeks) or aged (20 months) mice was transferred to young adolescent (5 weeks) germ-free (GF) mice (N = 5 per group) by fecal microbiota transplantation (FMT). Four weeks after the first FMT was performed, fecal samples were collected for 16S rRNA sequencing. Blood, heart and liver samples were harvested for oxidative stress marker and antioxidant defense analysis. The results showed that mice that received young or aged microbiota showed clear differences in GM composition and diversity. Mice that received aged microbiota had a lower ratio of Bacteroidetes/Firmicutes in GM at the phylum level and an increased relative abundance of four GM genera: *Akkermansia*, *Dubosiella*, *Alistipes* and *Rikenellaceae_RC9_gut_group*. In addition, GM α-diversity scores based on the Shannon index and Simpson index were significantly higher in aged GM-treated mice. Oxidative stress marker and antioxidant defense tests showed that FMT from aged donors did not have a significant influence on malondialdehyde content in serum, heart and liver. However, the capacity of anti-hydroxyl radicals in the heart and liver, as well as the capacity of anti-superoxide anions in the liver, were significantly increased in mice with aged microbiota. FMT from aged donors increased the activities of Cu/Zn superoxide SOD (Cu/Zn-SOD), catalase (CAT) and glutathione-*S*-transferase in the heart, as well as the activity of Cu/Zn-SOD in the liver. Positive correlations were found between Cu/Zn-SOD activity and radical scavenging capacities. On the other hand, glutathione reductase activity and glutathione content in the liver were decreased in mice that received aged GM. These findings suggest that aged GM transplantation from hosts is sufficient to influence the antioxidant defense

**Funding:** This research was financially supported by the Sichuan Provincial Science and Technology Plan Project (2022YFS0432), Basic Research Project of Sichuan Province (2022JDKY0013), Scientific and Technical Research Project of Sichuan Provincial Administration of TCM (2023MS485). The funders had no role in study design, data collection and analysis, decision to publish, or preparation of the manuscript.

**Competing interests:** The authors have declared that no competing interests exist.

system of young adolescent recipients in an organ-dependent manner, which highlights the importance of the GM in the aging process of the host.

## Introduction

Over recent decades, life expectancy has significantly increased globally. However, this increase has not led to a corresponding improvement in health outcomes, which is represented by an increase in numerous age-related diseases. It is well accepted that oxidative stress is an important contributor to aging, and a state of chronic oxidative stress characterizes older organisms and participates pivotally in the development of aging-related diseases [1]. Oxidative stress is the result of an elevation of reactive oxygen species (ROS), which are produced mainly in mitochondria as byproducts of normal cell respiration [1]. To remove excessive ROS, aerobic organisms have developed antioxidant defenses, which mainly rely on antioxidants, such as antioxidant enzymes and nonenzymatic glutathione (GSH) [1]. However, this critical defense has been found to decline with increasing aging, which is regarded as one major cause of the generation of oxidative stress in elderly individuals [1]. Hence, factors that undermine the antioxidant defense in elderly individuals need to be identified, which would be highly beneficial in developing interventions to promote healthy aging.

The human gut microbiota (GM), comprised of approximately $10^{14}$ microorganisms, is a complex and dynamic population that resides in the gastrointestinal tract [2]. To date, the interactions between the GM and host aging are a hotspot of research interest and have shed new light on the involvement of the GM in aging-associated decreases in antioxidant defense [3]. Generally, the composition of GM changes with aging, which leads to great GM differences between young and elderly subjects [4]. Recently, using a D-galactose-induced mouse aging model, studies showed that the alleviation of oxidative stress in aging bodies was associated with restoration of GM structure [3, 5–7]. In addition, Brunt et al. (2019) reported that suppression of the GM by antibiotics ameliorated age-related vascular oxidative stress and antioxidant enzyme decreases in mice [8]. To date, there are few studies concerning the direct effect of an aged GM on the antioxidant defense of the host. Colonizing young pseudo germ-free (GF) rats with the GM of aged rats by fecal microbiota transplantation (FMT), Li et al. (2020) reported that aged GM induced superoxide dismutase (SOD) enzyme activity decrease and malondialdehyde (MDA) elevation in the serum, hippocampus and medial prefrontal cortex [9]. Based on the findings above, an aged GM might modulate antioxidant defense, thus inducing oxidative stress in the host. If this is true, it would provide important evidence to explore strategies targeting GM-related alleviation in age-related oxidative stress.

As the organs requiring a high-energy source generated from mitochondrial respiration, the heart and liver are particularly prone to suffer from oxidative stress [1]. Indeed, a substantial percentage of morbidity and mortality in the elderly is attributable to oxidative stress-associated heart and liver diseases, which also cause tremendous clinical and economic burden outcomes [10, 11]. On the other hand, studies have shown that there are distinct variations in vulnerability to oxidative stress among different organs and tissues [12]. Postmitotic tissues, such as the heart, are more prone to oxidative stress than mitotic tissues (e.g., liver) due to their reduced ability to upregulate antioxidant defenses and/or repair accumulated oxidative damage [12]. Compared with the liver, a greater increase in oxidative stress has been found in the heart of aging rats [13]. We speculated that an aged GM might generate greater antioxidant defense impairment, thus leading to a higher level of oxidative stress in the heart than in the liver. To test this hypothesis, in this study, we conducted FMT from aged and young

adolescent donor mice into young adolescent GF mice. Then, the impact of the aged GM on oxidative stress and antioxidant defense in the heart and liver of recipient mice was explored. FMT is the delivery of a donor microbiota to a recipient in the form of a purified fecal microbiota or a stool suspension [14]. Combined with the use of GF animals, FMT has become one of the most common techniques to analyze the effect of different types of GM on the physiological state of hosts [14]. As mentioned above, the present study aimed to explore the influences of an aged GM on the antioxidant defense of the host.

## Materials and methods

All animal experiments were carried out according to ethical policies and procedures approved by the Animal Care Advisory Committee of the Sichuan Academy of Chinese Medicine Sciences (SYLL (2022)-047).

### Mice

C57BL/6-GF mice (male; 4 weeks) were purchased from the Third Military Medical University (Chongqing, China). Upon arrival, the mice were transferred into flexible film isolators in the gnotobiotic facility at the Sichuan Academy of Chinese Medicine Sciences. Fecal samples were collected from each mouse to ensure the absence of bacteria in the gut according to Lee et al. (2020) [15]. After one week of acclimation, mice were exported from flexible film isolators into sterile individually ventilated cages (IVCs) for fecal transplantation [15–17]. We included only males in this study for the following reasons: (1) There are differences in the GM and antioxidant response between males and females, and the gender effect is not the subject of our research [18, 19]; (2) according to the animal welfare guidelines of our institution, a single sex is suggested for a single study.

### Fecal microbiota transplantation (FMT)

Donors for FMT were specific pathogen-free C57BL/6 male mice (5 weeks & 20 months; N = 10 per group). Young adolescent recipients were GF C57BL/6 male mice aged 5 weeks (N = 5 per group). Fresh fecal samples were collected from the donors, and all fecal samples from the same group were mixed, homogenized in ice-cold PBS (120 mg feces/1 ml) and centrifuged at $800 \times g$ for 3 min at 4 ˚C [15]. Then, 100 μL of this supernatant was given to recipient mice by oral gavage on Days 1, 4, 7, 14, and 21 [9, 15, 20]. Recipients were individually housed in IVCs for 4 weeks [21]. The mice gavaged with fecal supernatant from aged C57BL/6 mice were referred to as the aged-FMT group, and the other group was referred to as the young-FMT group.

### 16S rRNA sequencing

Feces were collected from all recipient mice at the end of the experiment and immediately frozen at -80˚C. DNA isolation and sequencing, as well as data analysis, were carried out at Novogene Co., Ltd. (China). Approximately 100 mg of fecal samples were used for total genome DNA extraction using TIANGEN kits. By performing 1% agarose gel electrophoresis, DNA concentration and purity were assessed.

The extracted DNA fragments were subjected to Illumina MiSeq deep sequencing. Briefly, amplification of the V4 hypervariable region of the 16S rRNA gene was accomplished by using the 515F (5′ GTGCCAGCMGCCGCGGTAA 3′) and 806R (5′ GGACTACVSGGGTATCTAAT 3′) set of primers with a barcode. PCR was performed using Phusion® High-FIdelity PCR Master Mix (New England Biolabs, Ipswich, MA) with GC buffer with the reaction conditions

described by Frolinger et al. [22]. Then, the PCR products were run on agarose gel electrophoresis, purified using the TIANGEN Gel Extraction kit (TIANGEN, China) and sequenced by the Illumina NovaSeq6000 platform (Novogene Bioinformatics Technology Co., Ltd., Tianjin, China). According to the manufacturer's instructions, sequencing library generation was accomplished by using the NEBNext Ultra DNA Library Prep Kit for Illumina (Illumina, USA), and the index codes were assigned. The Illumina paired-end reads were overlapped and merged using Flash ver. 1.2.7 after cutting off the primer sequence and the barcode. Then, high-quality clean tags were obtained according to Bokulich et al. [23]. Subsequently, chimera filtering was undertaken using VSEARCH. Chimeric sequences were removed according to Hess et al. [24] to obtain effective tags.

Based on 97% sequence similarity, the generation of the operational taxonomic unit (OTU) table was achieved by using UPARSE v.7.0.1001 software [25]. Then, a representative sequence was chosen for each OTU, annotating the taxonomic information of that unit. Subsequently, all OTUs were analyzed for abundance and diversity. The analysis of alpha (α) diversity was calculated using Mothur. A UPGMA tree was constructed on the basis of the UniFrac distance calculated by using QIIME (Version 1.9.1).

## Reduction–oxidation (Redox) state analysis

Mice were sacrificed at the end of the experiment. Mice were anesthetized with isoflurane inhalation, bled and then sacrificed by cervical dislocation. Then, the blood, heart and liver were collected. Serum was prepared by centrifuging the blood samples at *3 000 ×g* for 15 min. All the samples were immediately frozen and stored at -80 ˚C until analysis. Heart and liver samples were homogenized on ice in 10 volumes (w/v) of ice-cold physiological saline and centrifuged at *6000 g* for 20 min at 4 ˚C, and then supernatants were collected for analysis. The MDA content in the serum, heart and liver was investigated using the thiobarbituric acid reaction, which was accomplished by using a test kit (Nanjing Jiancheng Bioengineer Institute, Nanjing, China). Briefly, serum or supernatant samples were mixed with trichloroacetic acid and centrifuged. Subsequently, thiobarbituric acid was added to the supernatant. The mixture underwent a 40-minute heating process in water at 95 ˚C, resulting in the formation of a red complex between MDA and thiobarbituric acid, the absorption of which was measured by spectrophotometry at 532 nm. The Fenton reaction ($Fe^{2+}+H_2O_2–Fe^{3+}+OH^-+OH^•$) was used to evaluate the anti-hydroxyl radical (AHR) capacity by using a test kit (Nanjing Jiancheng Bioengineer Institute, Nanjing, China), in which hydroxyl radicals ($OH^•$) are generated. Upon addition of the electron acceptor, a coloration reaction (absorbance at 550 nm) is induced by utilizing the Griess reagent. The degree of coloration is directly proportional to the concentration of hydroxyl radicals. The anti-superoxide anion (ASA) capacity was analyzed following the instructions of the test kit (Nanjing Jiancheng Bioengineer Institute, Nanjing, China). In brief, superoxide radicals ($O^{2•−}$) were generated through the action of xanthine and xanthine oxidase. When an electron acceptor is introduced, a coloration reaction is initiated using the Griess reagent. The intensity of the coloration (absorbance at 550 nm) is directly proportional to the amount of superoxide anion present in the reaction. In addition, the activities of antioxidant enzymes, including Cu/Zn superoxide dismutase (Cu/Zn-SOD) and Mn superoxide dismutase (Mn-SOD), were assessed as described by Jiang et al. [26]. The reaction mixture consisted of 30 μL of tissue sample supernatant, 1.08 mmol $L^{-1}$ diethylenetriaminepentaacetic acid, 50 mmol $L^{-1}$ phosphate buffer (pH 7–8), 0.16 mmol $L^{-1}$ xanthine solution, and 0.06 mmol $L^{-1}$ nitro blue tetrazolium. Following the introduction of 0.19 U $mL^{-1}$ xanthine oxidase, an alteration in absorbance at 550 nm was observed. To distinguish between SOD isoforms, the responsiveness of Cu/Zn-SOD to cyanide was employed, while Mn-SOD remained

unaffected. The decomposition of hydrogen peroxide was determined to analyze catalase (CAT) activity [26]. The assay mixture comprised a total volume of 1 mL containing 100 mmol L$^{-1}$ KPO$_4$ buffer (pH 7.0), 10 mmol L$^{-1}$ H$_2$O$_2$, and 50 μL of tissue supernatant. The reduction of H$_2$O$_2$ was analyzed by measuring the absorbance at 240 nm. Glutathione peroxidase (GPX) activity was detected according to the method of Jiang et al. (2012). In brief, the reaction mixture consisted of 15 μL of tissue supernatant, 40 μL of 0.25 mmol L$^{-1}$ H$_2$O$_2$, 0.5 mmol L$^{-1}$ GSH, 10 mmol L$^{-1}$ sodium phosphate buffer (pH 7.0) and 1.25 mmol L$^{-1}$ NaN$_3$ in a total volume of 1 mL. After 3-min intervals, 0.5 mL of dithiobisnitrobenzoic acid was introduced. The formation of a yellow product, resulting from the reaction between GSH and dithiobisnitrobenzoic acid, was observed and measured at 412 nm. The activity of glutathione reductase (GR) was tested by measuring the rate of NADPH oxidation [27], and the main procedure was as follows: The initial reaction mixture contained 1 mM EDTA, 0.1 mM NADPH, 100 mM NaCl/P$_i$ (pH 7.0), and 100 μL of tissue sample supernatant in a final volume of 1 mL. In this assay, the oxidation of NADPH, regardless of GR, was assessed by measuring the reduction in absorbance at 340 nm over a period of 60 seconds. Subsequently, 50 μL of 20 mM GSSG was added, and the reduction in absorbance at 340 nm was observed for an additional 120 seconds. Glutathione-*S*-transferase (GST) activity was analyzed according to the method of Lushchak et al. [28]. The reaction mixture consisted of 5 mmol L$^{-1}$ GSH, 50 mmol L$^{-1}$ KPi buffer (pH 7.0), 1 mmol L$^{-1}$ 1-chloro-2,4-dinitrobenzene (CDNB), 0.5 mmol L$^{-1}$ EDTA, and 40 μL of tissue sample supernatant. The formation of an adduct between GSH and CDNB was observed and measured at 340 nm. The determination of GSH content was accomplished by the formation of 5-thio-2-nitrobenzoate, which was subsequently measured spectrophotometrically at a wavelength of 412 nm [29].

## Statistical analysis

The redox state parameter results were expressed as the means±SDs. The differences between the two groups were analyzed with a two-tailed t test. Differences in the α-diversity of the GM between the two groups were analyzed with the Wilcox test. The correlations between antioxidant enzyme activity and radical scavenging capacity were analyzed using Pearson's correlation analysis. P<0.05 was considered significant, and P<0.01 was considered very significant.

## Results

A total of 86,487 tags were obtained from 10 samples, which were clustered into 1051 OTUs. To determine the extent of similarity in gut microbial communities between the groups, UniFrac distance was calculated. The UPGMA tree built using weighted UniFrac distances revealed clustering of aged-FMT group fecal samples from young-FMT group fecal samples (Fig 1). The phylum composition of GM in each group is shown in Fig 2A. In agreement with previous results, Firmicutes and Bacteroidetes phyla covered the vast majority of the microbiota [30], and the ratio of Bacteroidetes/Firmicutes was decreased in the aged-FMT group. The GM composition was also analyzed at the genus level. The relative abundances of the genera in the two groups are shown in Fig 2B. Among all the top 10 genera, the relative abundance of the genera *Akkermansia*, *Dubosiella*, *Alistipes* and *Rikenellaceae_RC9_gut_group* was increased in the aged-FMT group. For GM diversity, α-diversity scores based on the Shannon index and Simpson index were significantly higher in the aged-FMT group (Fig 3).

The MDA content in serum, heart and liver was not significantly different between the groups (Fig 4). As shown in Fig 4, the capacity of AHR in the heart and liver, as well as the capacity of ASA in the liver, were significantly increased in the aged-FMT group. In contrast, the capacity of AHR in serum, as well as the capacity of ASA in serum and heart, did not

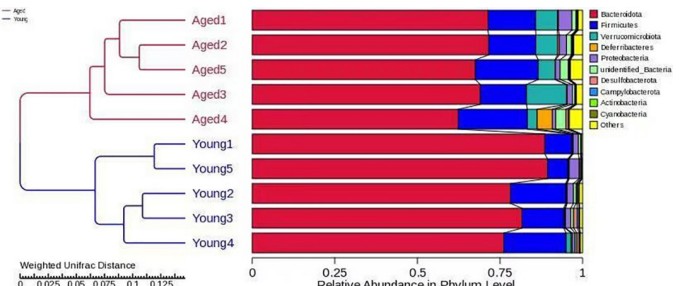

**Fig 1. UPGMA clustering analysis of gut microbiota based on weighted UniFrac distance.** The left side displays the structure of the clustering tree, while the right side shows the relative abundance of different phyla.

significantly differ between the groups. Fig 5 reveals the antioxidant indicators. FMT from aged donors increased the activities of Cu/Zn-SOD, CAT and GST in the heart, as well as the activity of Cu/Zn-SOD in the liver of the recipient mice. However, the activity of GR in the liver was significantly decreased in the aged-FMT group. In addition, the activities of Mn-SOD, GPX and GR in the heart, as well as the activities of Mn-SOD, CAT, GPX and GST in the liver, were not significantly different between the groups. FMT from aged donors did not have a significant effect on GSH content in the heart but decreased GSH content in the liver of the recipient mice (Fig 5). To uncover the potential relationship between the increased antioxidant enzyme activities and the increased radical scavenging capacities, a Pearson correlation analysis was conducted. The results only showed positive correlations between Cu/Zn-SOD activity and radical scavenging capacities (Table 1).

## Discussion

In recent years, the close relationship between the GM and host aging has become evident. However, the influence of age-associated shifts in the GM on host antioxidant defense remains to be determined. Antioxidant defense impairment is a major contributor to age-related oxidative stress, which is associated with various diseases in elderly individuals [1]. Here, we attempted to uncover the alterations in oxidative stress and antioxidant defense in the heart

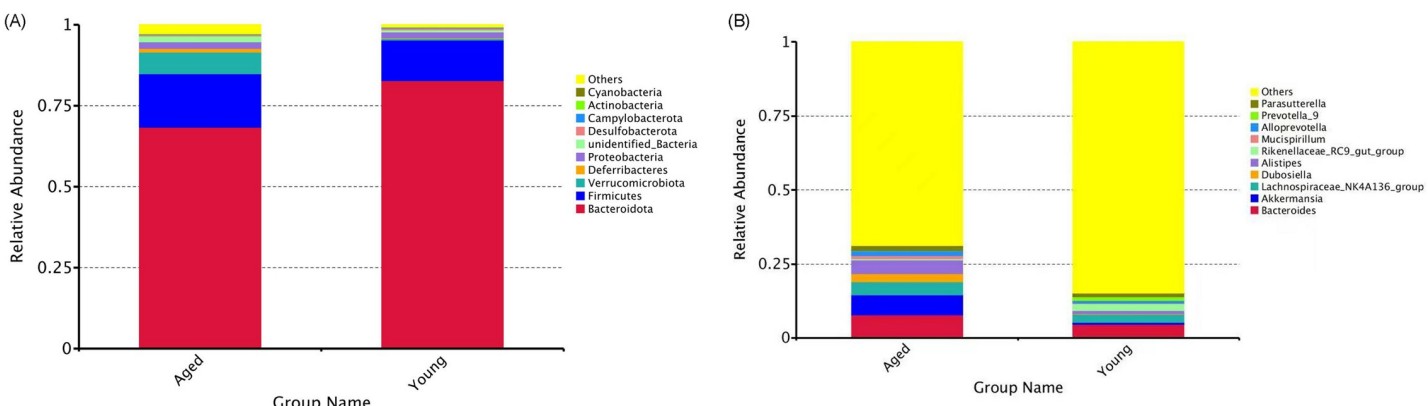

**Fig 2. Gut microbiota composition at the level of phylum and genus.** (A) Relative abundance of gut microbiota at the phylum level. (B) Relative abundance of gut microbiota at the genus level. Each bar represents the relative abundance of a bacterial taxon, with the top 10 most abundant taxa being displayed.

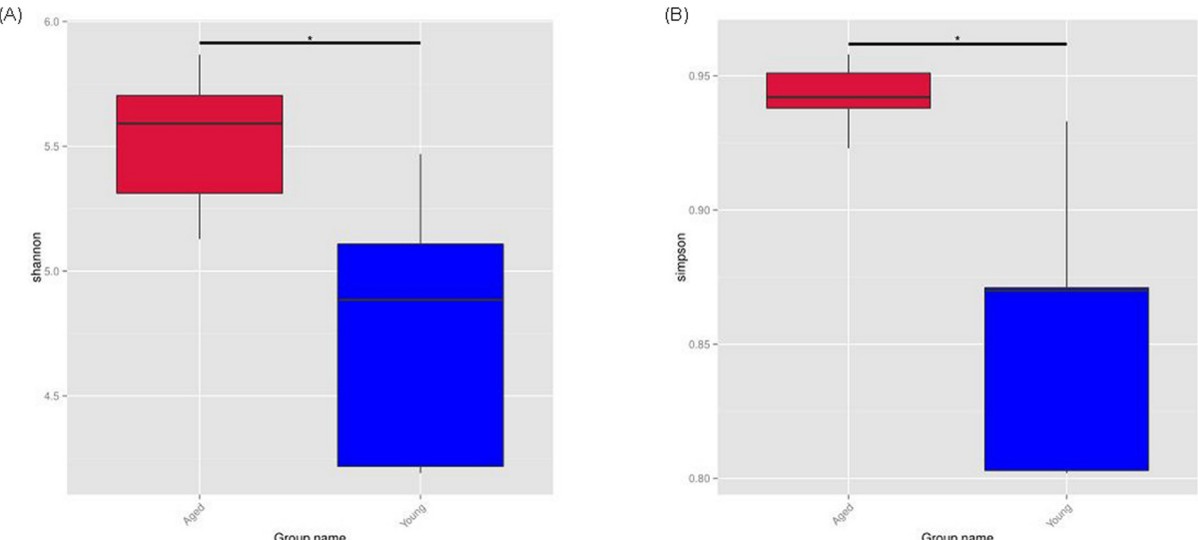

**Fig 3. α-diversity of gut microbiota.** (A) Shannon index. (B) Simpson index. The bottom and top boundaries of each box show the 25th and 75th percentiles, respectively. The line within each box represents the median. Ends of the whiskers indicate the maximum and minimum values. * indicates P<0.05.

and liver of young adolescent GF mice that received GM from either young or aged adolescent donor mice. First, the GM profiles between the young and aged recipients were compared to confirm successful GM transfer. After FMT for 4 weeks, significant differences existed between the GM of the aged-FMT group and the young-FMT group, as revealed by the UPGMA clustering result. Our study showed that both the composition and diversity of the GM differed between the two groups. *Firmicutes* and *Bacteroides* are two dominant gut bacteria at the phylum level. In this study, a decreased ratio of *Firmicutes* to *Bacteroides* (F/B) was found in the aged-FMT group, which was consistent with a previous result [9]. Moreover, intergroup GM variability was also observed at the genus level. Compared with the young-FMT group, the relative abundance of the genera *Akkermansia*, *Dubosiella*, *Alistipes* and *Rikenellaceae_RC9_-gut_group* was increased in the aged-FMT group. Among those four GM genera, *Alistipes* and *Rikenellaceae_RC9_gut_group* were found to be positively associated with age and some diseases [31, 32]. However, *Akkermansia* and *Dubosiella* were indicated to act as healthy microbiota in the aging process [33, 34]. Regarding diversity, the GM of mice in the aged-FMT group had higher Shannon and Simpson indices, reflecting higher species richness and evenness [33]. This result was contrary to our expectations, as a decrease in GM diversity with aging has been widely reported [35–37]. However, some studies have also shown an increase in GM diversity with aging in long-living people [38]. Together with the increased *Akkermansia* and *Dubosiella* in the aged-FMT group, we speculated that the aged donor mice used in this study might represent a healthy aging group.

Oxidative stress is a major factor involved in the pathogenesis of age-associated diseases. With increasing age, oxidative stress tends to increase in a GM-associated manner [5–7]. The index typically used to assess the degree of oxidative stress is the product of lipid peroxidation, named MDA [1]. In this study, FMT from aged donors failed to induce MDA elevation in the serum, heart and liver of the recipients, and all the MDA contents were similar to those observed in mice under normal physiological conditions [39, 40], which indicates that there is no obvious oxidative stress in the recipient mice. To date, information regarding the influence

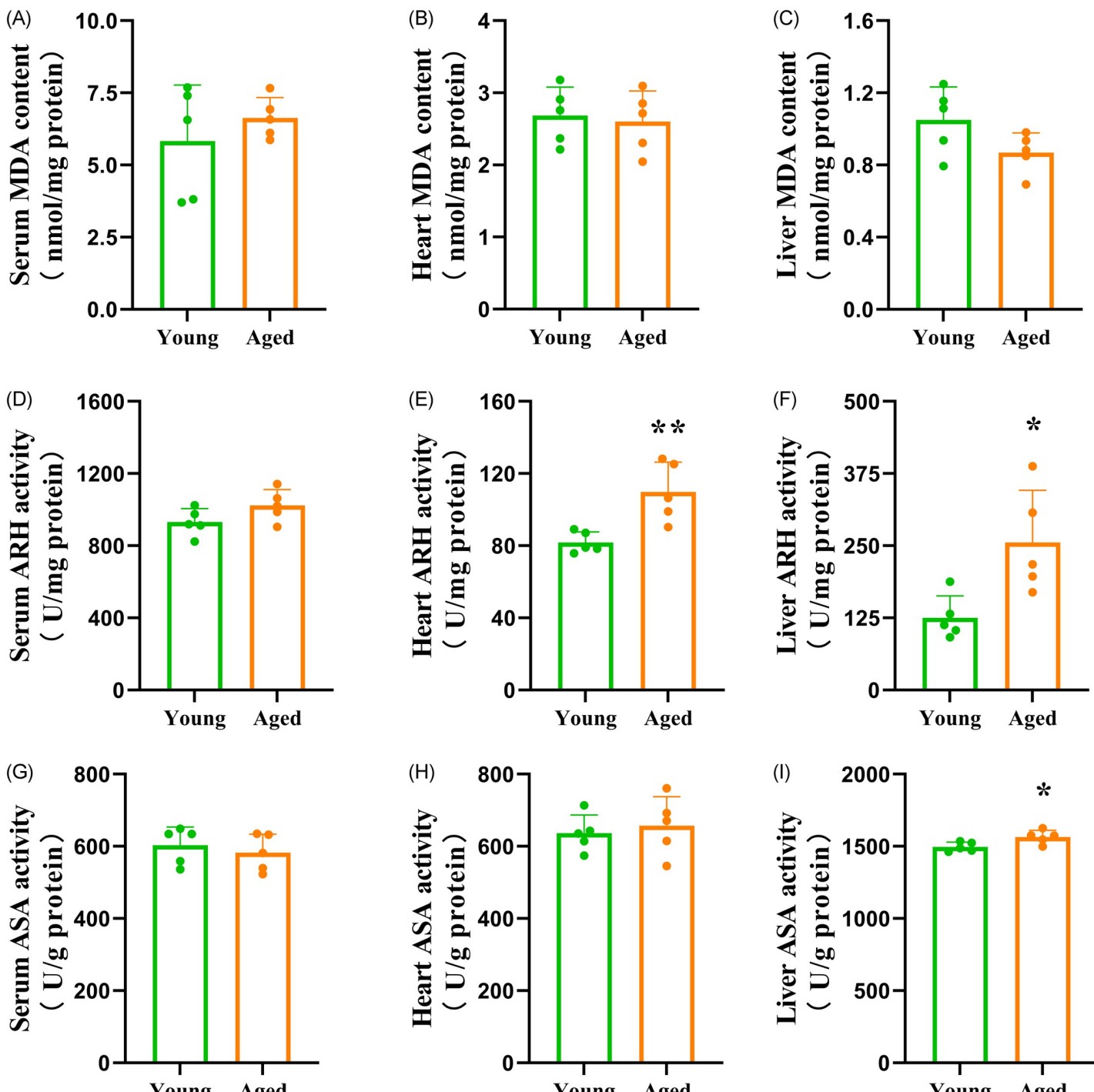

**Fig 4. Levels of MDA, AHR and ASA in the serum, heart and liver of the recipient mice.** The bars represent the mean±SD (n = 5). * indicates P<0.05, and ** represents P<0.01. (A) Serum MDA content = serum malondialdehyde content; (B) Heart MDA content = heart malondialdehyde content; (C) Liver MDA content = liver malondialdehyde content; (D) Serum AHR = serum anti-hydroxyl radical capacity; (E) Heart AHR = heart anti-hydroxyl radical capacity; (F) Liver AHR = liver anti-hydroxyl radical capacity; (G) Serum ASA = serum anti-superoxide anion capacity; (H) Heart ASA = heart anti-superoxide anion capacity; (I) Liver ASA = liver anti-superoxide anion capacity.

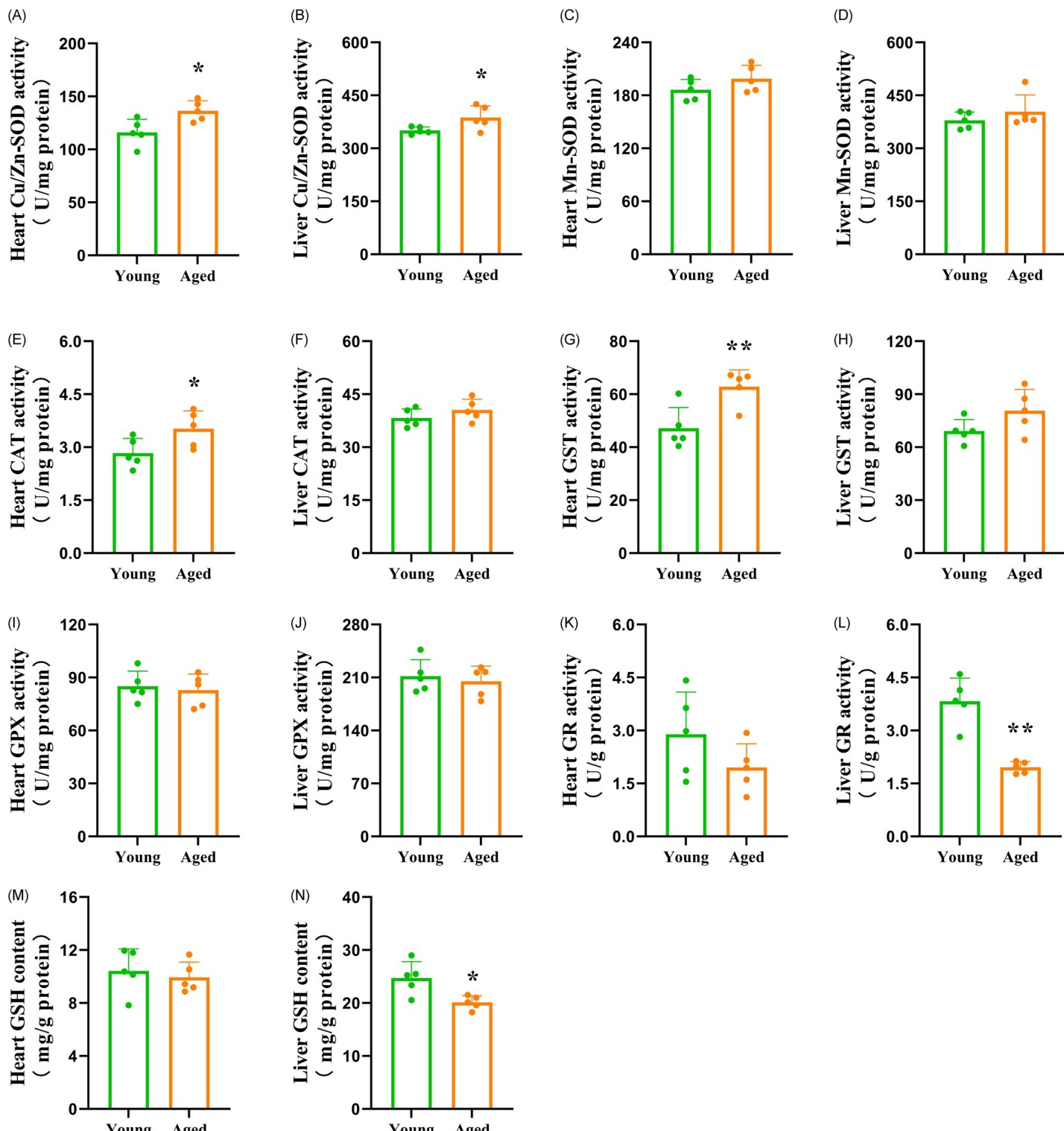

**Fig 5. Activities of antioxidant enzymes and GSH content in the heart and liver of recipient mice.** The bars represent the mean±SD (n = 5). * indicates P<0.05, and ** represents P<0.01. (A) Heart Cu/Zn-SOD activity = heart Cu/Zn superoxide dismutase activity; (B) Liver Cu/Zn-SOD activity = liver Cu/Zn superoxide dismutase activity; (C) Heart Mn-SOD activity = heart Mn superoxide dismutase activity; (D) Liver Mn-SOD activity = liver Mn superoxide dismutase activity; (E) Heart CAT activity = heart catalase activity; (F) Liver CAT activity = liver catalase activity; (G) Heart GST activity = heart glutathione-*S*-transferase activity; (H) Liver GST activity = liver glutathione-*S*-transferase activity; (I) Heart GPX activity = heart glutathione peroxidase activity; (J) Liver GPX activity = liver glutathione peroxidase activity; (K) Heart GR activity = heart glutathione reductase activity; (L) Liver GR activity = liver glutathione reductase activity; (M) Heart GSH content = heart glutathione content; (N) Liver GSH content = liver glutathione content.

**Table 1. Correlations between Cu/Zn-SOD activity and radical scavenging capacity.**

| Antioxidant parameters | Correlation | Radical scavenging capacity |
|---|---|---|
| **Heart Cu/Zn-SOD activity (U/mg protein)** | + | Heart AHR (r = 0.693 p = 0.026) |
| **Liver Cu/Zn-SOD activity (U/mg protein)** | + | Liver AHR (r = 0.867 p = 0.01) |
| **Liver Cu/Zn-SOD activity (U/mg protein)** | + | Liver ASA (r = 0.886 p = 0.01) |

Cu/Zn-SOD: Cu/Zn superoxide dismutase; AHR: anti-hydroxyl radical capacity; ASA: anti-superoxide anion capacity; +: positive correlation; r: Pearson's correlation coefficient

of aged GM on oxidative stress in GF recipient mice is limited. Li et al. [9] reported that an aged GM induced elevation of serum and brain MDA content in young pseudo-GF rats. Such discrepancies might be correlated with differences in GM characteristics. At the genus level, aged FMT induced a totally different microbiota composition change in pseudo-GF rats compared with our study. In addition, the results of GM diversity change were also completely opposite. As healthy and unhealthy aging subjects have great differences in GM and physiological characteristics, our study, associated with the report of Li et al. [9], highlights the importance of considering GM characteristics in studying topics of aging-related oxidative status. Despite the insignificant difference in MDA content between the recipients, our further test showed some changes in the antioxidant system induced by aged-FMT. $OH^{\bullet}$ and $O^{2\bullet-}$ are two free radicals that are highly involved in oxidative stress [1]. $OH^{\bullet}$ is considered the most reactive oxygen radical in biological systems and can react rapidly with many molecules in the cell, including proteins, lipids, DNA, and others [1]. $O^{2\bullet-}$ can cause great oxidative damage within the cell due to its relatively long half-life [1]. Two indices, namely, antihydroxyl radical (AHR) capacity and antisuperoxide anion (ASA) capacity, are used to represent the total scavenging ability against $OH^{\bullet}$ and $O^{2\bullet-}$, respectively. In this study, we observed different responses of these two capacities to aged GM between the heart and liver, which was a proof that the heart has a reduced ability to upregulate antioxidant defense compared with the liver. Transplantation of aged-GM did not have a significant influence on ASA capacity in the heart. However, AHR capacity in the heart, as well as ASA and AHR capacity in the liver, were significantly higher in the aged-FMT group. As the MDA content was similar between the young and aged-FMT groups, the enhanced free radical scavenging ability indicates a response against increased ROS production in the aged-FMT group [41]. Studies have shown that some gut bacterial metabolites, such as small formylated peptides and lipopolysaccharides, could contribute to host ROS production [42, 43], and those metabolites were found to increase with increasing host age [44]. However, further studies are needed to test the influence of aged GM on host ROS production.

In this study, the levels of antioxidant compounds were analyzed to further determine the possible mechanism by which aged-FMT enhanced the free radical scavenging abilities in young adolescent GF recipients. Endogenous antioxidant enzymes provide the first line of cellular defense against toxic free radicals [1]. SOD, a metalloenzyme, is the first enzyme involved in responding to oxygen free radicals [1]. On the basis of the type of metal ion required as cofactor, various forms of SOD exist. Cu/Zn-SOD is widely distributed in the cytoplasm of eukaryotes, while Mn-SOD is present in mitochondria [1]. In this study, the activity of Mn-SOD was not significantly different between the groups. However, the activity of Cu/Zn-SOD in the heart and liver was improved in the aged-FMT group. CAT is essential for the detoxification of hydroxyl radicals [1]. In this study, CAT activity in the heart was higher in the aged-FMT group. Furthermore, glutathione-dependent enzymes are also able to protect cells against

free radical-induced injury [1]. Our study showed that FMT from aged donors enhanced the activity of GST in the heart but had no significant effect on the activities of GPX and GR in the heart or the activities of GPX and GST in the liver. On the other hand, FMT from aged donors decreased liver GR activity in recipient mice. It is well known that GR is capable of catalyzing the reduction of oxidized GSH [1]. Similar to the GR activity, the GSH content in the liver was lower in the aged-FMT group, which might be partly attributed to the decrease in GR activity. Among all antioxidant compounds mentioned above, Pearson's correlation analysis uncovered positive correlations between the Cu/Zn-SOD activity and the radical scavenging capacities, which indicated that the increased radical scavenging capacities were partially attributed to the increased Cu/Zn-SOD activity. Liu et al. (2022) reported that *Dubosiella newyorkensis*, a strain of *Dubosiella*, possessed a strong effect of increasing SOD in aged mice [34]. Thus, in this study, the increased *Dubosiella* might play an important role in shaping SOD activity, as well as radical scavenging capacity characteristics in the heart and liver. Further study is needed to test this hypothesis. Comparing the antioxidant defense changes between the heart and liver, it is evident that an aged GM had varying effects on antioxidant defense in these organs. It has been widely reported that antioxidant defense follows a mixed pattern of age-related changes, and those changes are markedly variable depending on the tissues and organs examined [18, 45–47]. Our study indicated that GM is involved in these tissue- and organ-dependent antioxidant defense changes during aging.

In conclusion, the present study highlights the importance of the GM in aging-related antioxidant defense changes in the host. Transplantation of an aged GM into young adolescent GF mice did not have an impact on lipid peroxidation but enhanced free radical scavenging abilities in the heart and liver. The aged GM had different influences on antioxidant enzyme activities and GSH content between the heart and liver, which suggests an organ-dependent effect of the aged GM on host antioxidant defense. Among all the significantly changed antioxidant compounds, Cu/Zn-SOD activity was a factor that contributed to the enhanced radical scavenging abilities in aged GM recipients. Our study identified significant differences in four types of GM genera (*Akkermansia*, *Dubosiella*, *Alistipes* and *Rikenellaceae_RC9_gut_group*) between the two groups. Further work is warranted to investigate the detailed mechanisms by which the GM affects the antioxidant response in elderly individuals.

## Author Contributions

**Conceptualization:** Ya Luo.

**Data curation:** Yang-Lu Liu.

**Investigation:** Jie-Ying Xia.

**Methodology:** Ya Luo.

**Software:** Jing Zhou.

**Supervision:** Lei Zhang.

**Validation:** Ming-Ming Yuan.

**Visualization:** Jia-Fei Zhan.

**Writing – original draft:** Han Dong.

**Writing – review & editing:** Yang Hong.

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
