## [Decision Letter · Decision Letter 0]

2 Jun 2023

PONE-D-23-09106Aged Versus Young Gut Microbiota Transplants to Young Germ-free C57BL/6 Mice: Changes in the Reduction-Oxidation (Redox) State in the Heart and LiverPLOS ONE

Dear Dr. Zhang,

Thank you for submitting your manuscript to PLOS ONE. After careful consideration, we feel that it has merit but does not fully meet PLOS ONE’s publication criteria as it currently stands. Therefore, we invite you to submit a revised version of the manuscript that addresses the points raised during the review process.

Both reviewers request experimental clarifications, rationale for some study design, and major revisions of text. Please note that a revised manuscript will be sent back to the same reviewers.

We look forward to receiving your revised manuscript.

Kind regards,

David M. Ojcius

Academic Editor

PLOS ONE

Journal Requirements:

"This research was supported by the Basic Research Project of Sichuan Province (2022JDKY0013)."

7. Your ethics statement should only appear in the Methods section of your manuscript. If your ethics statement is written in any section besides the Methods, please move it to the Methods section and delete it from any other section. Please ensure that your ethics statement is included in your manuscript, as the ethics statement entered into the online submission form will not be published alongside your manuscript. 

Reviewers' comments:

Reviewer's Responses to Questions

**Comments to the Author**

1. Is the manuscript technically sound, and do the data support the conclusions?

Reviewer #1: Partly

Reviewer #2: Yes

2. Has the statistical analysis been performed appropriately and rigorously? 

Reviewer #1: Yes

Reviewer #2: I Don't Know

3. Have the authors made all data underlying the findings in their manuscript fully available?

Reviewer #1: Yes

Reviewer #2: No

4. Is the manuscript presented in an intelligible fashion and written in standard English?

Reviewer #1: No

Reviewer #2: Yes

5. Review Comments to the Author

Reviewer #1: Aged Versus Young Gut Microbiota Transplants to Young Germ-free C57BL/6 Mice: Changes in the Reduction-Oxidation (Redox) State in the Heart and Liver

The authors present an interesting manuscript looking at FMT of aged microbiome into you and aged germ free mice.

There are several issues with the manuscript that will require a lot of work.

Major

Why was aged microbiome used as FMT material? There is no rationale or translational validity to this experiment that I can see.

There is no clear hypothesis that I can see.

The study is underpowered, was a power calculation performed?

The grammar throughout needs revision.

Show graphs rather than tables for the redox experiments.

Include individual data points.

The authors need to discuss why they looked at the heart and the liver, there is not rationale introduced in the introduction.

Please give a rationale for why only male mice were used.

Were the mice germ free on purchase?

What was the rationale for the time of administration of the FMT, was it administered at the same time every day?

line 91, OUT spelt wrong.

What was the statistical method for analysing microbiome?

Line 98 is confusing, 4th week of what?

How was the serum compared?

More details of the methods for redox analysis are needed

Reviewer #2: The study investigates the effects of gut microbiota on the redox state in two important organs such as heart and liver. In particular, they used FMT mouse-to-mouse approach using two different types of donors: aged and young. The recipients of the FMT were young germ-free mice. They evaluated the engraftment of the gut microbiota after the FMT and also the redox state in heart and liver. The work is interesting. However, the paper could be improved doing some major corrections. Taken together, the paper could be acceptable for publication only if the comments are addressed, as needs a big change. Please read my comments:

1. Title: In my opinion, the title is too long. For example, use only Germ-Free and indicate that you use C57BL/6 in the body of the paper. Choose between Reduction-Oxidation or Redox, as sound redundant. Perhaps, you can create a more informative title rather than a descriptive title.

2. Abstract: The abstract needs more work as is not clear on the objectives, the methods and the results are not clear. For example, in the line 9 the acronym GM is not clear if refers to age gut microbiota or just gut microbiota, so this makes a bit confusing the rest of the reading. Another example is that the reason of why the study was performed is not clear.

3. Introduction:

a. Lines 32-34, this refers to FMT studies? What are the characteristics of the aging in rat brain and mice intestine?

b. Line 49: Why is important or interesting to evaluate Redox state in heart and liver?

c. Line 53: As the term reduction-oxidation has been associated with the acronym redox, use that throughout the manuscript, as can be confusing to read it all the time.

4. Material and Methods:

a. Mice: Just wondering why only males were used in the study? Is there a prevalence in alteration of the redox state in a male population rather than the females? You purchase the mice at 4 weeks old and then performed the experiments at 5 weeks old, which to my understanding correspond to adolescence period, so you might clarify that you are doing and intervention in the adolescence and the readouts are in adulthood. The term “young” is not clear and is important to mention that the intervention was performed during adolescence as many processes are still on going.

b. FMT: Is the FMT procedure performed as one-to-one mouse?

c. Redox state analysis: Can the isoflurane influence the redox state in the tissue analysed? Performing only cervical dislocation was not an option? In general, this section needs more attention, a brief explanation on how each analysis was performed would improve the paper.

5. Results:

a. Figures: Please provide a more elaborated figure legend for ach figure.

b. Figure 3: Elaborate the legend in terms of what * means, what are the error bars SD or SEM? Add more info to the letters used like A) Shannon, B) Simpson. This will help the reader to better understand the data.

c. Redox data: Perhaps having the data in graphs instead of table might help the reader to visualise the findings. As there are quite a lot of results, this can be clustered in panels.

In general, the figures need to be made in a better image quality. Ignore these comments if the image quality was affected during the PDF exportation process.

6. Discussion: The begging of the discussion need rephrasing as is confusing.

a. Line 167: What about the MDA levels in GF mice? Is this parameter already altered in the model? Please add discussion around this.

b. Line 172: Please elaborate on how and what future work should be performed.

c. Line 174: O2 and OH terms has not been used before so can be confusing to the reader to interpret the provided discussion. Please clarify this throughout the manuscript.

d. Line 179: Please elaborate regarding the importance or relevance of the increased rate of free radicals in the aged-FMT, what this mean?

e. Lines 198-200: Is the gut microbiota involved in this process? In general, there is a very little information about the gut microbiota in the manuscript and why this type of intervention was chosen. In the introduction should be more info about the microbiota and why the FMT approach was used.

6. PLOS authors have the option to publish the peer review history of their article (what does this mean?). If published, this will include your full peer review and any attached files.

Reviewer #1: **Yes: **Gerard Moloney

Reviewer #2: No

---

## [Author Response · Author response to Decision Letter 0]

19 Jul 2023

Response to reviewer #1

Dear reviewer,

Thank you very much for your review and kind suggestions. Also, we take great appreciation for your kindness that giving us an opportunity to revise. We have seriously revised our manuscript according to your advising. The detail responses list as follows.

1: Why was aged microbiome used as FMT material? There is no rationale or translational validity to this experiment that I can see.

Answer: Thank you very much for your kind suggestion. According to your comment, the reason that an aged microbiota was used as FMT material has been added in Instruction, page 4, line 54-68:

“To date, the interactions between the GM and host aging are a hotspot of research interest and have shed new light on the involvement of the GM in aging-associated decreases in antioxidant defense [3]. Generally, the composition of GM changes with aging, which leads to great GM differences between young and elderly subjects [4]. Recently, using a D-galactose-induced mouse aging model, studies showed that the alleviation of oxidative stress in aging bodies was associated with restoration of GM structure [3, 5-7]. In addition, Brunt et al. (2019) reported that suppression of the GM by antibiotics ameliorated age-related vascular oxidative stress and antioxidant enzyme decreases in mice [8]. To date, there are few studies concerning the direct effect of an aged GM on the antioxidant defense of the host. Colonizing young pseudogerm-free (GF) rats with the GM of aged rats by fecal microbiota transplantation (FMT), Li et al. (2020) reported that aged GM induced superoxide dismutase (SOD) enzyme activity decrease and malondialdehyde (MDA) elevation in the serum, hippocampus and medial prefrontal cortex [9]. Based on the findings above, an aged GM might modulate antioxidant defense, thus inducing oxidative stress in the host. If this is true, it would provide important evidence to explore strategies targeting GM-related alleviation in age-related oxidative stress. ”

2: There is no clear hypothesis that I can see.

Answer: Thank you very much for your kind suggestion. According to your comment, we have added some hypothesizes in the Instruction. The detail changes are listed as follows.

Instruction, page 4, line 66-67:

The hypothesis that an aged microbiota could influence the antioxidant defense and oxidative stress of the host has been added: “Based on the findings above, an aged GM might modulate antioxidant defense, thus inducing oxidative stress in the host.”

Instruction, page 4, line 77-79:

The hypothesis that an aged microbiota might induce different influences on antioxidant defense and oxidative stress between the heart and liver has been added: “We speculated that an aged GM might generate greater antioxidant defense impairment, thus leading to a higher level of oxidative stress in the heart than in the liver.”

3. The study is underpowered, was a power calculation performed?

Answer: Thanks very much for your kind review. We very appreciate your precious suggestion. According to your advice, we have added some calculations. The detail changes are listed as follows.

The data of gut microbiota composition at genus level has been added in Results, page 10, line 201-204:“The GM composition was also analyzed at the genus level. The relative abundances of the genera in the two groups are shown in Fig. 2B. Among all the top 10 genera, the relative abundance of the genera Akkermansia, Dubosiella, Alistipes and Rikenellaceae_RC9_gut_group was increased in the aged-FMT group.”

The correlations between increased antioxidant enzymes and increased radical scavenging capacities were analyzed and results have been added (Results, page 11, line 231-234): “To uncover the potential relationship between the increased antioxidant enzyme activities and the increased radical scavenging capacities, a Pearson correlation analysis was conducted. The results only showed positive correlations between Cu/Zn-SOD activity and radical scavenging capacities (Table 1).”

4. The grammar throughout needs revision.

Answer: Thanks very much for your kind review. According to your advice, our manuscript has been edited by the American Journal Experts (AJE).

5. Show graphs rather than tables for the redox experiments.

Answer: Thank you very much for your kind suggestion. According to your comment, we have presented the redox data in the form of graphs.

6. Include individual data points.

Answer: Thank you very much for your kind comments. According to your suggestion, the individual data points have been added in graphs.

7. The authors need to discuss why they looked at the heart and the liver, there is not rationale introduced in the introduction.

Answer: Thank you very much for your kind comments. According to your suggestion, the reason that the heart and liver were investigated has been added in Instruction, page 4, line 69-79: “As the organs requiring a high-energy source generated from mitochondrial respiration, the heart and liver are particularly prone to suffer from oxidative stress [1]. Indeed, a substantial percentage of morbidity and mortality in the elderly is attributable to oxidative stress-associated heart and liver diseases, which also cause tremendous clinical and economic burden outcomes [10,11]. On the other hand, studies have shown that there are distinct variations in vulnerability to oxidative stress among different organs and tissues [12]. Postmitotic tissues, such as the heart, are more prone to oxidative stress than mitotic tissues (e.g., liver) due to their reduced ability to upregulate antioxidant defenses and/or repair accumulated oxidative damage [12]. Compared with the liver, a greater increase in oxidative stress has been found in the heart of aging rats [13]. We speculated that an aged GM might generate greater antioxidant defense impairment, thus leading to a higher level of oxidative stress in the heart than in the liver.”

8. Please give a rationale for why only male mice were used.

Answer: Thank you very much for your kind comments. According to your suggestion, the reason that only male mice were used has been added in Materials and Methods, page 5, line 97-101: “We included only males in this study for the following reasons: (1) There are differences in the GM and antioxidant response between males and females, and the gender effect is not the subject of our research [18,19]; (2) according to the animal welfare guidelines of our institution, a single sex is suggested for a single study.”

9. Were the mice germ free on purchase?

Answer: The germ free mice were purchased, and the information was provided in Materials and Methods, page 5, line 92-93: “C57BL/6-GF mice (male; 4 weeks) were purchased from the Third Military Medical University (Chongqing, China).”

10. What was the rationale for the time of administration of the FMT, was it administered at the same time every day?

Answer: Thanks very much for your kind review. The timing of fecal microbiota transplantation (FMT) is based on methods used in other studies. Currently, there exists some variability in the timing of FMT across different studies; however, they generally adhere to the following pattern: during the first week, it is more frequent, around 3 times. From the second week onwards, it is done once or twice a week. The exact time for FMT administration was not mentioned in most reported studies. We conducted FMT at 8:00 am according to Wen et al. (2021).

11. line 91, OUT spelt wrong.

Answer: Thanks very much for your kind review. We were very sorry for the wrong spelt, and the word has been corrected accordingly. 

Materials and Methods, page 7, line 131:

“OUT” has been changed to “OTU”.

12. What was the statistical method for analysing microbiome?

Answer: Thanks very much for your kind review. According to your comment, the statistical method for analysing microbiome was added in Statistical Analysis, page 9, line 190-191: “Differences in the α-diversity of the GM between the two groups were analyzed with the Wilcox test.”

13. Line 98 is confusing, 4th week of what?

Answer: Thanks very much for your kind review. According to your comment, the sentence “Mice were sacrificed at the end of forth week” have been changed to “Mice were sacrificed at the end of the experiment.” in Materials and Methods, page 7, line 138.

14. How was the serum prepared?

Answer: Thanks very much for your kind review. According to your comment, the method for serum preparation has been added in Materials and Methods, page 7, line 140: “Serum was prepared by centrifuging the blood samples at 3 000 ×g for 15 min.”

15. More details of the methods for redox analysis are needed.

Answer: Thank you very much for your kind suggestion. According to your comment, the details of the methods for redox analysis have been added in Materials and Methods, page 7-9, line 143-187: “The MDA content in the serum, heart and liver was investigated using the thiobarbituric acid reaction, which was accomplished by using a test kit (Nanjing Jiancheng Bioengineer Institute, Nanjing, China). Briefly, serum or supernatant samples were mixed with trichloroacetic acid and centrifuged. Subsequently, thiobarbituric acid was added to the supernatant. The mixture underwent a 40-minute heating process in water at 95 ℃, resulting in the formation of a red complex between MDA and thiobarbituric acid, the absorption of which was measured by spectrophotometry at 532 nm. The Fenton reaction (Fe2++H2O2－Fe3++OH-+OH•) was used to evaluate the anti-hydroxyl radical (AHR) capacity by using a test kit (Nanjing Jiancheng Bioengineer Institute, Nanjing, China), in which hydroxyl radicals (OH•) are generated. Upon addition of the electron acceptor, a coloration reaction (absorbance at 550 nm) is induced by utilizing the Griess reagent. The degree of coloration is directly proportional to the concentration of hydroxyl radicals. The anti-superoxide anion (ASA) capacity was analyzed following the instructions of the test kit (Nanjing Jiancheng Bioengineer Institute, Nanjing, China). In brief, superoxide radicals (O2•−) were generated through the action of xanthine and xanthine oxidase. When an electron acceptor is introduced, a coloration reaction is initiated using the Griess reagent. The intensity of the coloration (absorbance at 550 nm) is directly proportional to the amount of superoxide anion present in the reaction. In addition, the activities of antioxidant enzymes, including Cu/Zn superoxide dismutase (Cu/Zn-SOD) and Mn superoxide dismutase (Mn-SOD), were assessed as described by Jiang et al. (2012) [26]. The reaction mixture consisted of 30 µL of tissue sample supernatant, 1.08 mmol L-1 diethylenetriaminepentaacetic acid, 50 mmol L-1 phosphate buffer (pH 7-8), 0.16 mmol L-1 xanthine solution, and 0.06 mmol L-1 nitro blue tetrazolium. Following the introduction of 0.19 U mL-1 xanthine oxidase, an alteration in absorbance at 550 nm was observed. To distinguish between SOD isoforms, the responsiveness of Cu/Zn-SOD to cyanide was employed, while Mn-SOD remained unaffected. The decomposition of hydrogen peroxide was determined to analyze catalase (CAT) activity [26]. The assay mixture comprised a total volume of 1 mL containing 100 mmol L-1 KPO4 buffer (pH 7.0), 10 mmol L-1 H2O2, and 50 μL of tissue supernatant. The reduction of H2O2 was analyzed by measuring the absorbance at 240 nm. Glutathione peroxidase (GPX) activity was detected according to the method of Jiang et al. (2012). In brief, the reaction mixture consisted of 15 μL of tissue supernatant, 40 μL of 0.25 mmol L-1 H2O2, 0.5 mmol L-1 GSH, 10 mmol L-1 sodium phosphate buffer (pH 7.0) and 1.25 mmol L-1 NaN3 in a total volume of 1 mL. After 3-min intervals, 0.5 mL of dithiobisnitrobenzoic acid was introduced. The formation of a yellow product, resulting from the reaction between GSH and dithiobisnitrobenzoic acid, was observed and measured at 412 nm. The activity of glutathione reductase (GR) was tested by measuring the rate of NADPH oxidation [27], and the main procedure was as follows: The initial reaction mixture contained 1 mM EDTA, 0.1 mM NADPH, 100 mM NaCI/Pi (pH 7.0), and 100 µL of tissue sample supernatant in a final volume of 1 mL. In this assay, the oxidation of NADPH, regardless of GR, was assessed by measuring the reduction in absorbance at 340 nm over a period of 60 seconds. Subsequently, 50 µL of 20 mM GSSG was added, and the reduction in absorbance at 340 nm was observed for an additional 120 seconds. Glutathione-S-transferase (GST) activity was analyzed according to the method of Lushchak et al. (2011) [28]. The reaction mixture consisted of 5 mmol L-1 GSH, 50 mmol L-1 KPi buffer (pH 7.0), 1 mmol L-1 1-chloro-2,4-dinitrobenzene (CDNB), 0.5 mmol L-1 EDTA, and 40 µL of tissue sample supernatant. The formation of an adduct between GSH and CDNB was observed and measured at 340 nm. The determination of GSH content was accomplished by the formation of 5-thio-2-nitrobenzoate, which was subsequently measured spectrophotometrically at a wavelength of 412 nm [29].”

Reference:

Wen, X., Wang, H. G., Zhang, M. N., Zhang, M. H., Wang, H., & Yang, X. Z. (2021). Fecal microbiota transplantation ameliorates experimental colitis via gut microbiota and T-cell modulation. World Journal of Gastroenterology, 27(21), 2834.

 

Response to reviewer #2

Dear reviewer,

Thank you very much for your review and kind suggestions. Also, we take great appreciation for your kindness that giving us an opportunity to revise. We have seriously revised our manuscript according to your advising. The detail responses list as follows.

1. Title: In my opinion, the title is too long. For example, use only Germ-Free and indicate that you use C57BL/6 in the body of the paper. Choose between Reduction-Oxidation or Redox, as sound redundant. Perhaps, you can create a more informative title rather than a descriptive title.

Answer: Thank you very much for your kind suggestion. According to your comment, the tile “Aged Versus Young Gut Microbiota Transplants to Young Germ-free C57BL/6 Mice: Changes in the Reduction-Oxidation (Redox) State in the Heart and Liver” has been changed to “Aged Gut Microbiota Contribute to Different Changes in Antioxidant Defense in the Heart and Liver after Transfer to Germ-free Mice”.

2. Abstract: The abstract needs more work as is not clear on the objectives, the methods and the results are not clear. For example, in the line 9 the acronym GM is not clear if refers to age gut microbiota or just gut microbiota, so this makes a bit confusing the rest of the reading. Another example is that the reason of why the study was performed is not clear.

Answer: Thanks very much for your kind review. According to your suggestion, we have revised the abstract carefully. The final version of abstract is as follows.

Age-associated impairment in antioxidant defense is an important cause of oxidative stress, and elderly individuals are usually associated with gut microbiota (GM) changes. Studies have suggested a potential relationship between the GM and changes in antioxidant defense in aging animals. Direct evidence regarding the impact of aging-associated shifts in GM on the antioxidant defense is lacking. The heart is a kind of postmitotic tissue, which is more prone to oxidative stress than the liver (mitotic tissue). To test and compare the influence of an aged GM on antioxidant defense changes in the heart and liver of the host, in this study, GM from young adolescent (5 weeks) or aged (20 months) mice was transferred to young adolescent (5 weeks) germ-free (GF) mice (N=5 per group) by fecal microbiota transplantation (FMT). Four weeks after the first FMT was performed, fecal samples were collected for 16S rRNA sequencing. Blood, heart and liver samples were harvested for oxidative stress marker and antioxidant defense analysis. The results showed that mice that received aged or young microbiota showed clear differences in GM composition and diversity. Mice that received aged microbiota had a lower ratio of Bacteroidetes/Firmicutes in GM at the phylum level and an increased relative abundance of four GM genera: Akkermansia, Dubosiella, Alistipes and Rikenellaceae_RC9_gut_group. In addition, GM α-diversity scores based on the Shannon index and Simpson index were significantly higher in aged GM-treated mice. Oxidative stress marker and antioxidant defense tests showed that FMT from aged donors did not have a significant influence on malondialdehyde content in serum, heart and liver. However, the capacity of anti-hydroxyl radicals in the heart and liver, as well as the capacity of anti-superoxide anions in the liver, were significantly increased in mice with aged microbiota. FMT from aged donors increased the activities of Cu/Zn superoxide SOD (Cu/Zn-SOD), catalase (CAT) and glutathione-S-transferase in the heart, as well as the activity of Cu/Zn SOD in the liver. Positive correlations were found between Cu/Zn-SOD activity and radical scavenging capacities. On the other hand, glutathione reductase activity and glutathione content in the liver were decreased in mice that received aged GM. These findings suggest that aged GM transplantation from hosts is sufficient to influence the antioxidant defense system of young adolescent recipients in an organ-dependent manner, which highlights the importance of the GM in the aging process of the host.

3. a. Lines 32-34, this refers to FMT studies? What are the characteristics of the aging in rat brain and mice intestine?

Answer: Thank you very much for your kind review. Both of these two studies have used FMT method to investigate effects of aged GM on aging characteristics in rat brain and mouse intestine (Li et al. 2020; Fransen et al. 2017). Li et al. (2020) showed that the aged GM FMT increased pro-inflammatory cytokines and oxidative stress (increased MDA content, decreased SOD activity) in the brain of young pseudo-germ free rats. Fransen et al. (2017) reported that aged GM promoted inflammation in the small intestine of GF mice. Due to the revision of our manuscript, the information of Fransen et al. (2017) was removed from our manuscript, and the information of Li et al. (2020) has been revised to: “Colonizing young pseudogerm-free (GF) rats with the GM of aged rats by fecal microbiota transplantation (FMT), Li et al. (2020) reported that aged GM induced superoxide dismutase (SOD) enzyme activity decrease and malondialdehyde (MDA) elevation in the serum, hippocampus and medial prefrontal cortex” in Introduction, page 4, line 63-66.

4. Line 49: Why is important or interesting to evaluate Redox state in heart and liver?

Answer: Thank you very much for your kind comments. According to your suggestion, the reason that the heart and liver were investigated has been added in Instruction, page 4, line 69-79: “As the organs requiring a high-energy source generated from mitochondrial respiration, the heart and liver are particularly prone to suffer from oxidative stress [1]. Indeed, a substantial percentage of morbidity and mortality in the elderly is attributable to oxidative stress-associated heart and liver diseases, which also cause tremendous clinical and economic burden outcomes [10, 11]. On the other hand, studies have shown that there are distinct variations in vulnerability to oxidative stress among different organs and tissues [12]. Postmitotic tissues, such as the heart, are more prone to oxidative stress than mitotic tissues (e.g., liver) due to their reduced ability to upregulate antioxidant defenses and/or repair accumulated oxidative damage [12]. Compared with the liver, a greater increase in oxidative stress has been found in the heart of aging rats [13]. We speculated that an aged GM might generate greater antioxidant defense impairment, thus leading to a higher level of oxidative stress in the heart than in the liver.”

5. Line 53: As the term reduction-oxidation has been associated with the acronym redox, use that throughout the manuscript, as can be confusing to read it all the time.

Answer: Thank you very much for your kind comments. Due to the revision of our manuscript, the term “reduction-oxidation” has only been used twice in the text. The second “reduction-oxidation” has been abbreviated as “redox” in Statistical Analysis, page 9, line 189.

6. Mice: Just wondering why only males were used in the study? Is there prevalence in alteration of the redox state in a male population rather than the females? You purchase the mice at 4 weeks old and then performed the experiments at 5 weeks old, which to my understanding correspond to adolescence period, so you might clarify that you are doing and intervention in the adolescence and the readouts are in adulthood. The term “young” is not clear and is important to mention that the intervention was performed during adolescence as many processes are still on going.

(1) Mice: Just wondering why only males were used in the study?

Answer: Thank you very much for your kind review. According to your comments, the reason that only male mice were used has been added in Materials and Methods, page 5, line 97-101: “We included only males in this study for the following reasons: (1) There are differences in the GM and antioxidant response between males and females, and the gender effect is not the subject of our research [18,19]; (2) according to the animal welfare guidelines of our institution, a single sex is suggested for a single study.”

(2) Is there prevalence in alteration of the redox state in a male population rather than the females?

Answer: Thank you very much for your kind review. Studies have demonstrated that aging had significant influences on redox state in both males and females. Besides, those influences were apparently sex-dependent. For example, Carrillo et al. (1992) found sex-dependent pattern of age-related changes of antioxidant enzymes in rat liver, including CAT, Cu/Zn-SOD and Mn-SOD (Table 1). In our study, we didn’t investigate the gender effect, which would be important and interesting topics in future studying.

Table 1. A sex-dependent pattern of age-related changes of antioxidant enzymes in rats

Antioxidant enzymes Gender Young rat Old rat

CAT male High activity Low activity

 female Low activity High activity

Cu/Zn SOD male Low activity High activity

 female No significant difference No significant difference

Mn SOD male High activity Low activity

 female Low activity High activity

(3) You purchase the mice at 4 weeks old and then performed the experiments at 5 weeks old, which to my understanding correspond to adolescence period, so you might clarify that you are doing and intervention in the adolescence and the readouts are in adulthood. The term “young” is not clear and is important to mention that the intervention was performed during adolescence as many processes are still on going.

Answer: Thank you very much for your kind suggestion. According to your comments, all the terms “young” have been changed to “young adolescent”. The change spots are as follows.

Abstract, page 2, line 18;

Abstract, page 2, line 19;

Abstract, page 3, line 37;

Introduction, page 4, line 80;

Materials and Methods, page 6, line 104;

Discussion, page 12, line 260-262;

Discussion, page 14, line 314;

Discussion, page 16, line 341.

7. FMT: Is the FMT procedure performed as one-to-one mouse?

Answer: Thank you very much for your kind review. The FMT procedure is performed according to the method reported by Lee et al. (2020), and it is not in a one-to-one way. We have added the information in Materials and Methods, page 6, line 105-106: “and all fecal samples from the same group were mixed”.

8. Redox state analysis: Can the isoflurane influence the redox state in the tissue analysed? Performing only cervical dislocation was not an option? In general, this section needs more attention, a brief explanation on how each analysis was performed would improve the paper.

(1) Can the isoflurane influence the redox state in the tissue analysed? Performing only cervical dislocation was not an option?

Answer: Thank you very much for your kind review. So far, performing ocervical dislocation with or without isoflurane is both used frequently in studies on redox state. We chose this method according to Tsai et al. (2018). Besides, the blood and tissue samples were collected very quickly after anesthesia, so redox state indexes should not be influenced. 

(2) A brief explanation on how each analysis was performed would improve the paper.

Answer: Thank you very much for your kind suggestion. According to your comments, the explanation on how each analysis was performed has been added in Materials and Methods, page 7-9, line 143-187: “The MDA content in the serum, heart and liver was investigated using the thiobarbituric acid reaction, which was accomplished by using a test kit (Nanjing Jiancheng Bioengineer Institute, Nanjing, China). Briefly, serum or supernatant samples were mixed with trichloroacetic acid and centrifuged. Subsequently, thiobarbituric acid was added to the supernatant. The mixture underwent a 40-minute heating process in water at 95 ℃, resulting in the formation of a red complex between MDA and thiobarbituric acid, the absorption of which was measured by spectrophotometry at 532 nm. The Fenton reaction (Fe2++H2O2－Fe3++OH-+OH•) was used to evaluate the anti-hydroxyl radical (AHR) capacity by using a test kit (Nanjing Jiancheng Bioengineer Institute, Nanjing, China), in which hydroxyl radicals (OH•) are generated. Upon addition of the electron acceptor, a coloration reaction (absorbance at 550 nm) is induced by utilizing the Griess reagent. The degree of coloration is directly proportional to the concentration of hydroxyl radicals. The anti-superoxide anion (ASA) capacity was analyzed following the instructions of the test kit (Nanjing Jiancheng Bioengineer Institute, Nanjing, China). In brief, superoxide radicals (O2•−) were generated through the action of xanthine and xanthine oxidase. When an electron acceptor is introduced, a coloration reaction is initiated using the Griess reagent. The intensity of the coloration (absorbance at 550 nm) is directly proportional to the amount of superoxide anion present in the reaction. In addition, the activities of antioxidant enzymes, including Cu/Zn superoxide dismutase (Cu/Zn-SOD) and Mn superoxide dismutase (Mn-SOD), were assessed as described by Jiang et al. (2012) [26]. The reaction mixture consisted of 30 µL of tissue sample supernatant, 1.08 mmol L-1 diethylenetriaminepentaacetic acid, 50 mmol L-1 phosphate buffer (pH 7-8), 0.16 mmol L-1 xanthine solution, and 0.06 mmol L-1 nitro blue tetrazolium. Following the introduction of 0.19 U mL-1 xanthine oxidase, an alteration in absorbance at 550 nm was observed. To distinguish between SOD isoforms, the responsiveness of Cu/Zn-SOD to cyanide was employed, while Mn-SOD remained unaffected. The decomposition of hydrogen peroxide was determined to analyze catalase (CAT) activity [26]. The assay mixture comprised a total volume of 1 mL containing 100 mmol L-1 KPO4 buffer (pH 7.0), 10 mmol L-1 H2O2, and 50 μL of tissue supernatant. The reduction of H2O2 was analyzed by measuring the absorbance at 240 nm. Glutathione peroxidase (GPX) activity was detected according to the method of Jiang et al. (2012). In brief, the reaction mixture consisted of 15 μL of tissue supernatant, 40 μL of 0.25 mmol L-1 H2O2, 0.5 mmol L-1 GSH, 10 mmol L-1 sodium phosphate buffer (pH 7.0) and 1.25 mmol L-1 NaN3 in a total volume of 1 mL. After 3-min intervals, 0.5 mL of dithiobisnitrobenzoic acid was introduced. The formation of a yellow product, resulting from the reaction between GSH and dithiobisnitrobenzoic acid, was observed and measured at 412 nm. The activity of glutathione reductase (GR) was tested by measuring the rate of NADPH oxidation [27], and the main procedure was as follows: The initial reaction mixture contained 1 mM EDTA, 0.1 mM NADPH, 100 mM NaCI/Pi (pH 7.0), and 100 µL of tissue sample supernatant in a final volume of 1 mL. In this assay, the oxidation of NADPH, regardless of GR, was assessed by measuring the reduction in absorbance at 340 nm over a period of 60 seconds. Subsequently, 50 µL of 20 mM GSSG was added, and the reduction in absorbance at 340 nm was observed for an additional 120 seconds. Glutathione-S-transferase (GST) activity was analyzed according to the method of Lushchak et al. (2011) [28]. The reaction mixture consisted of 5 mmol L-1 GSH, 50 mmol L-1 KPi buffer (pH 7.0), 1 mmol L-1 1-chloro-2,4-dinitrobenzene (CDNB), 0.5 mmol L-1 EDTA, and 40 µL of tissue sample supernatant. The formation of an adduct between GSH and CDNB was observed and measured at 340 nm. The determination of GSH content was accomplished by the formation of 5-thio-2-nitrobenzoate, which was subsequently measured spectrophotometrically at a wavelength of 412 nm [29].”

9. Figures: Please provide a more elaborated figure legend for each figure.

Answer: Thank you very much for your kind suggestion. According to your comment, we have provided more elaborated figure legends. They are as follows.

Results, page 10, line 211-212:

“The left side displays the structure of the clustering tree, while the right side shows the relative abundance of different phyla. ” has been added.

Results, page 10, line 213-216.

“Relative abundance of gut microbiota at the phylum level.” has been changed to “Gut microbiota composition at the level of phylum and genus (A) Relative abundance of gut microbiota at the phylum level. (B) Relative abundance of gut microbiota at the genus level. Each bar represents the relative abundance of a bacterial taxon, with the top 10 most abundant taxa being displayed.” 

Results, page 11, line 217-220.

“α-diversity of gut microbiota. (A) Shannon index. (B) Simpson index.” has been changed to “α-diversity of gut microbiota. (A) Shannon index. (B) Simpson index. The bottom and top boundaries of each box show the 25th and 75th percentiles, respectively. The line within each box represents the median. Ends of the whiskers indicate the maximum and minimum values. * indicates P<0.05.”

10. Figure 3: Elaborate the legend in terms of what * means, what are the error bars SD or SEM? Add more info to the letters used like A) Shannon, B) Simpson. This will help the reader to better understand the data.

Answer: Thank you very much for your kind suggestion. According to your comments, the explanation for Figure 3 has been added in Results, page 11, line 217-220: “The bottom and top boundaries of each box show the 25th and 75th percentiles, respectively. The line within each box represents the median. Ends of the whiskers indicate the maximum and minimum values. * indicates P<0.05.”

11. Redox data: Perhaps having the data in graphs instead of table might help the reader to visualise the findings. As there are quite a lot of results, this can be clustered in panels.

Answer: Thank you very much for your kind suggestion. According to your comment, we have presented the redox data in the form of graphs.

12. In general, the figures need to be made in a better image quality. Ignore these comments if the image quality was affected during the PDF exportation process. 

Answer: Thank you very much for your kind review. As you speculated, the figures quality was affected during the PDF exportation process. Thanks again.

13. Discussion: The beginning of the discussion need rephrasing as is confusing.

Answer: Thank you very much for your kind suggestion. According to your comment, we have rephrasing the beginning of the Discussion page 12, line 256-262: “In recent years, the close relationship between the GM and host aging has become evident. However, the influence of age-associated shifts in the GM on host antioxidant defense remains to be determined. Antioxidant defense impairment is a major contributor to age-related oxidative stress, which is associated with various diseases in elderly individuals [1]. Here, we attempted to uncover the alterations in oxidative stress and antioxidant defense in the heart and liver of young adolescent GF mice that received GM from either aged or young adolescent donor mice. First, the GM profiles between the young and aged recipients were compared to confirm successful GM transfer.”

14. Line 167: What about the MDA levels in GF mice? Is this parameter already altered in the model? Please add discussion around this.

Answer: Thank you very much for your kind review. We did not measure MDA levels in GF mice. However, values for MDA content in this study were similar to those observed in mice under normal physiological conditions. We have added the discussion around this in Discussion page 13, line 283-286: “In this study, FMT from aged donors failed to induce MDA elevation in the serum, heart and liver of the recipients, and all the MDA contents were similar to those observed in mice under normal physiological conditions [40, 41], which indicates that there is no obvious oxidative stress in the recipient mice.”

15. Line 172: Please elaborate on how and what future work should be performed.

Answer: Thank you very much for your kind suggestion. Due to the inclusion of data on GM composition at the genus level, we have discussed the reason for the discrepancy between the two studies. The details are as follows.

“Such discrepancies might be correlated with differences in GM characteristics. At the genus level, aged FMT induced a totally different microbiota composition change in pseudo-GF rats compared with our study. In addition, the results of GM diversity change were also completely opposite. As healthy and unhealthy aging subjects have great differences in GM and physiological characteristics, our study, associated with the report of Li et al. (2020) [9], highlights the importance of considering GM characteristics in studying topics of aging-related oxidative status.” has been added in Discussion page 13-14, line 288-294.

16. Line 174: O2 and OH terms has not been used before so can be confusing to the reader to interpret the provided discussion. Please clarify this throughout the manuscript.

Answer: Thank you very much for your kind suggestion. According to your comment, the interpretation on OH• and O2•− has been added in Discussion page 14, line 296-301: “OH• is considered the most reactive oxygen radical in biological systems and can react rapidly with many molecules in the cell, including proteins, lipids, DNA, and others [1]. O2•− can cause great oxidative damage within the cell due to its relatively long half-life [1]. Two indices, namely, antihydroxyl radical (AHR) capacity and antisuperoxide anion (ASA) capacity, are used to represent the total scavenging ability against OH• and O2•−, respectively. ”

17. Line 179: Please elaborate regarding the importance or relevance of the increased rate of free radicals in the aged-FMT, what this mean?

Answer: Thank you very much for your kind review. Oxidative stress is the result of an imbalance between ROS (including OH• and O2•−) production and clearance. An excessive of ROS induces lipid peroxidation, resulting in an elevation of MDA content. In our study, the ROS scavenging capacity was increased by aged-FM. If the level of ROS production is similar between the young and aged-FMT group, there should be a lower level of MDA content in aged-FMT group. However, intergoup MDA content was similar. Thus, it can be deduced that there was an elevation of ROS generation in aged-FMT group. Studies showed that some gut bacteria metabolites, such as small formylated peptides and lipopolysaccharides, could contribute to host ROS production, and those metabolites were found to increase with increasing host age. However, further studies are needed to test the influence of aged GM on host ROS production. Thanks again for your kind review.

18. Lines 198-200: Is the gut microbiota involved in this process? In general, there is a very little information about the gut microbiota in the manuscript and why this type of intervention was chosen. In the introduction should be more info about the microbiota and why the FMT approach was used.

(1) Lines 198-200: Is the gut microbiota involved in this process?

Answer: Thank you very much for your kind review. It has been widely reported that the antioxidant defense followed a mixed pattern of age-related changes, and those changes were markedly variable depending on tissues and organs examined. In our study, the GM is the independent variable. Thus, it can be concluded that GM is involved in this organ-dependent antioxidant defense change. The speculation has been added in Discussion page 15, line 338-339: “Our study indicated that GM is involved in these tissue- and organ-dependent antioxidant defense changes during aging.”

 (2) In general, there is a very little information about the gut microbiota in the manuscript and why this type of intervention was chosen. In the introduction should be more info about the microbiota and why the FMT approach was used.

Answer: Thank you very much for your kind suggestion. According to your comment, information about the microbiota has been added in the Introduction page 3-4, line 53-68: “The human gut microbiota (GM), comprised of approximately 1014 microorganisms, is a complex and dynamic population that resides in the gastrointestinal tract [2]. To date, the interactions between the GM and host aging are a hotspot of research interest and have shed new light on the involvement of the GM in aging-associated decreases in antioxidant defense [3]. Generally, the composition of GM changes with aging, which leads to great GM differences between young and elderly subjects [4]. Recently, using a D-galactose-induced mouse aging model, studies showed that the alleviation of oxidative stress in aging bodies was associated with restoration of GM structure [3, 5-7]. In addition, Brunt et al. (2019) reported that suppression of the GM by antibiotics ameliorated age-related vascular oxidative stress and antioxidant enzyme decreases in mice [8]. To date, there are few studies concerning the direct effect of an aged GM on the antioxidant defense of the host. Colonizing young pseudogerm-free (GF) rats with the GM of aged rats by fecal microbiota transplantation (FMT), Li et al. (2020) reported that aged GM induced superoxide dismutase (SOD) enzyme activity decrease and malondialdehyde (MDA) elevation in the serum, hippocampus and medial prefrontal cortex [9]. Based on the findings above, an aged GM might modulate antioxidant defense, thus inducing oxidative stress in the host. If this is true, it would provide important evidence to explore strategies targeting GM-related alleviation in age-related oxidative stress.”

Besides, information about why the FMT approach was used has been added in the Introduction page 4-5, line 82-85: “FMT is the delivery of a donor microbiota to a recipient in the form of a purified fecal microbiota or a stool suspension [14]. Combined with the use of GF animals, FMT has become one of the most common techniques to analyze the effect of different types of GM on the physiological state of hosts [14].”

Reference:

[1] Fransen F, van Beek AA, Borghuis T, Aidy SE, Hugenholtz F, van der Gaast-de Jongh C, et al. Aged gut microbiota contributes to systemical inflammaging after transfer to germ-free mice. Frontiers in immunology. 2017;8: 1385. 

[2] Li Y, Ning L, Yin Y, Wang R, Zhang Z, Hao L, et al. Age-related shifts in gut microbiota contribute to cognitive decline in aged rats. Aging. 2020;12: 7801-17.

[3] Carrillo, M. C., Kanai, S., Sato, Y., & Kitani, K. (1992). Age-related changes in antioxidant enzyme activities are region and organ, as well as sex, selective in the rat. Mechanisms of ageing and development, 65(2-3), 187-198.

[4] Tsai, M. S., Wang, Y. H., Lai, Y. Y., Tsou, H. K., Liou, G. G., Ko, J. L., & Wang, S. H. (2018). Kaempferol protects against propacetamol-induced acute liver injury through CYP2E1 inactivation, UGT1A1 activation, and attenuation of oxidative stress, inflammation and apoptosis in mice. Toxicology letters, 290, 97-109.

[5] Lee J, Venna VR, Durgan DJ, Shi H, Hudobenko J, Putluri N, et al. Young versus aged microbiota transplants to germ-free mice: Increased short-chain fatty acids and improved cognitive performance. Gut microbes. 2020;12: 1-14.

---

## [Decision Letter · Decision Letter 1]

28 Jul 2023

Aged Gut Microbiota Contribute to Different Changes in Antioxidant Defense in the Heart and Liver after Transfer to Germ-free Mice

PONE-D-23-09106R1

Dear Dr. Zhang,

We’re pleased to inform you that your manuscript has been judged scientifically suitable for publication and will be formally accepted for publication once it meets all outstanding technical requirements.

Kind regards,

David M. Ojcius

Academic Editor

PLOS ONE

Reviewers' comments:

Reviewer's Responses to Questions

**Comments to the Author**

1. If the authors have adequately addressed your comments raised in a previous round of review and you feel that this manuscript is now acceptable for publication, you may indicate that here to bypass the “Comments to the Author” section, enter your conflict of interest statement in the “Confidential to Editor” section, and submit your "Accept" recommendation.

Reviewer #1: All comments have been addressed

Reviewer #2: All comments have been addressed

2. Is the manuscript technically sound, and do the data support the conclusions?

Reviewer #1: Yes

Reviewer #2: Yes

3. Has the statistical analysis been performed appropriately and rigorously? 

Reviewer #1: Yes

Reviewer #2: Yes

4. Have the authors made all data underlying the findings in their manuscript fully available?

Reviewer #1: Yes

Reviewer #2: Yes

5. Is the manuscript presented in an intelligible fashion and written in standard English?

Reviewer #1: Yes

Reviewer #2: Yes

6. Review Comments to the Author

Reviewer #1: The authors have done a very good job of anwering all of the questions I had. Once the amendments are finalised, this paper is ready for publication.

Reviewer #2: The authors of the manuscript titled "Aged Gut Microbiota Contribute to Different Changes in Antioxidant Defense in the Heart and Liver after Transfer to Germ-free Mice" addressed all my comments. I believe the paper is good to be published.

7. PLOS authors have the option to publish the peer review history of their article (what does this mean?). If published, this will include your full peer review and any attached files.

Reviewer #1: **Yes: **Gerard Moloney

Reviewer #2: No

---

## [Editor Report · Acceptance letter]

1 Aug 2023

PONE-D-23-09106R1 

Aged Gut Microbiota Contribute to Different Changes in Antioxidant Defense in the Heart and Liver after Transfer to Germ-free Mice 

Dear Dr. Zhang:

I'm pleased to inform you that your manuscript has been deemed suitable for publication in PLOS ONE. Congratulations! Your manuscript is now with our production department. 

Kind regards, 

on behalf of

Dr. David M. Ojcius 

Academic Editor

PLOS ONE